# Physical Exercise and Occupational Therapy at Home to Improve the Quality of Life in Subjects Affected by Rheumatoid Arthritis: A Randomized Controlled Trial

**DOI:** 10.3390/healthcare11152123

**Published:** 2023-07-25

**Authors:** Dario Cerasola, Christiano Argano, Valeria Chiovaro, Tatjana Trivic, Tijana Scepanovic, Patrik Drid, Salvatore Corrao

**Affiliations:** 1Department of Psychology, Educational Science and Human Movement, University of Palermo, 90128 Palermo, Italy; dario.cerasola@unipa.it; 2Department of Internal Medicine IGR, National Relevance and High Specialization Hospital Trust ARNAS Civico, Di Cristina, Benfratelli, 90127 Palermo, Italy; chiovarovaleria@libero.it (V.C.); salvatore.corrao@unipa.it (S.C.); 3Faculty of Sport and Physical Education, University of Novi Sad, 21000 Novi Sad, Serbia; ttrivic@yahoo.com (T.T.); tijanascepanovic021@gmail.com (T.S.); patrikdrid@gmail.com (P.D.); 4Dipartimento di Promozione della Salute, Materno Infantile, Medicina Interna e Specialistica di Eccellenza “G. D’Alessandro” (PROMISE), University of Palermo, 90127 Palermo, Italy

**Keywords:** rheumatoid arthritis, occupational therapy, physical exercise, quality of life

## Abstract

Background: Rheumatoid arthritis (RA) is a chronic autoimmune inflammatory disease that affects synovial membranes and typically causes joint pain and swelling. The resulting disability of RA is due to the erosion of cartilage and bone from the inflamed synovial tissue. Occupational therapy is a strategy and technique to minimize the joints’ fatigue and effort. At the same time, physical exercise reduces the impact of systemic manifestations and improves symptoms in RA. This study investigates the role of a 30-day joint economy intervention (integration of physical exercise and occupational therapy) at home on the quality of life of subjects with RA. Methods: One hundred and sixty outpatients with RA were enrolled in a single-center trial with PROBE design and were divided into the intervention group (IG), which combined joint protection movements and physical exercise to maintain muscle tone at home, and the control group (CG). Both groups included 80 patients. In all patients, data from the disease activity score (DAS 28), health assessment questionnaire (HAQ), and short-form health survey (SF-12) “Italian version” were collected. In addition, to IG, a brochure was distributed, and the joint economy was explained, while to CG, the brochure only was distributed. The comparison between groups was made using Fisher’s exact test for contingency tables and the z-test for the comparison of proportions. The non-parametric Mann–Whitney U test was used to compare quantitative variables between groups. The Wilcoxon signed-ranked test was used for post-intervention versus baseline comparisons. Results: Among the recruited patients, 54% were female. The mean age was 58.0 (42.4–74.7) for the CG and 54.0 (39.7–68.3) for the IG. Patients included in the IG had a higher cumulative illness rating scale for the evaluation of severity and comorbidity index (2.81 vs. 2.58; 2.91 vs. 2.59, respectively), as well as morning stiffness (33.8 vs. 25.0), even if not significant compared with CG patients. Our results indicate that, after 30 days of joint economy intervention at home, the DAS28 erythrocyte sedimentation rate (esr) and DAS28 C-reactive protein (crp), HAQ, and SF-12 mental component score were significantly improved (*p* = 0.005, *p* = 0.004, *p* = 0.009, and *p* = 0.010, respectively). Conclusions: Our findings show that the combination of physical exercise and occupational therapy positively affects patients’ quality of life with RA considering disease activity, global health status, and mental health.

## 1. Introduction

Rheumatoid arthritis (RA) is a multi-factor chronic autoimmune inflammatory disease that primarily affects the lining of synovial joints and leads to the erosion of cartilage and bone [1]. It is one of the most severe chronic diseases, affecting about 0.5–1% of the general adult population with regional variation. Rheumatoid arthritis (RA) is two- to three-fold more frequent in women than in men [2]. It is characterized by painful symptoms, which result in a gradual loss of mobility and worsens quality of life due to the progressive destruction of the joint structures [3]. Frequently, RA presents different comorbidities, such as cardiovascular disease, depression, asthma, solid-organ malignancies, chronic obstructive pulmonary disease [4,5], and changes in body composition with decreased muscle mass and increased fat mass [6], representing a complex clinical picture [7]. As a consequence, subjects affected by RA have a working disability higher than the general population, with a negative influence on social, working, and environmental functioning [8,9]. For this reason, it requires a diagnostic approach and early intervention [10]. In this sense, the patient with RA is a complex patient that must be submitted to a multidisciplinary approach as in other diseases [11,12,13,14].

In patients with RA, joints, tendon sheaths, tendons, and ligaments affected by inflammation become more vulnerable to mechanical solicitations. Hence, the main objective of the treatment must be to suppress the activity of the disease and prevent structural and functional damage [15]. To achieve this goal, pharmacological and nonpharmacological therapeutic strategies are applied [16]. Nonpharmacological treatments include exercise, physical activity, self-management, and occupational therapy (OT) [17].

OT is defined as a set of strategies and techniques aimed at minimizing the efforts and fatigue of the joints and can therefore be considered as a set of tips aimed at learning the correct gestures that prevent and slow down the deterioration of the joints [18,19,20]. OT aims to help people overcome disabilities by facilitating the performance of activities of daily living, maintaining or improving abilities, or compensating for diminished abilities. Physical activity has positive effects on bone, on the one hand, slowing radiographic disease progression in small joints and increasing bone mineral density; on the other hand, it decreases comorbidity risk, pain perception, and symptoms of depression [21,22]. Moreover, it reduces the overall cost of the disease and improves self-movement [3]. Physical exercise programs include improved cardiorespiratory fitness and cardiovascular health, increased muscle mass, reduced adiposity (including attenuated trunk fat), improved strength, and physical functioning, which are all achieved without exacerbating disease activity or joint damage [23,24].

The integration of physical exercise and occupational therapy is named joint education, which allows patients with joint involvement to conduct normal daily activities and impart a real gesture of education, economizing on diseased joints to prevent joint damage.

Previous studies have shown that physical exercises and joint protection strategies positively improve social participation, deformities, pain reduction, fatigue, and functionality, and cause a reduction in morning stiffness in subjects with rheumatoid arthritis [25,26,27,28,29,30,31,32]. Moreover, a recent systematic review by Siegel and colleagues supported the use of patient education, self-management, multidisciplinary approaches, and joint protection in patients with rheumatoid arthritis [33]. In this sense, it has been hypothesized that a period of finalized exercises self-managed at home and habits can improve quality of life.

Given this background, this study aims to evaluate the quality of life in a sample of patients with rheumatoid arthritis after one month of physical exercise and occupational therapy intervention performed at home.

## 2. Materials and Methods

### 2.1. Subjects and Procedure

A single-center PROBE trial (Figure 1) was conducted in the Internal Medicine Department of ARNAS Civico-Di Cristina-Benfratelli Hospital of Palermo (Italy). A total of 160 consecutive outpatients with rheumatoid arthritis attending the Rheumatologic ambulatory were enrolled (86 females and 74 men). We completed the recruitment period within three months. After randomization, we took another three months to complete the follow-up. The reporting of the study followed the guidelines of the Consolidated Reporting of Trials (CONSORT). The present study was developed in accordance with the Declaration of Helsinki and had approval from the Ethics Committee of Palermo 2 with approval number 18-2021. All patients provided informed consent.

### 2.2. Selection and Randomization

The selection criteria included a diagnosis of rheumatoid arthritis, aged ≥ 18 years or older. Patients aged ≤ 18 and subjects with formed deformations of joints that prevent physical exercise or severe anemia were excluded. Eligible patients were randomly assigned to interventions using a computer-generated randomization sequence with a block size of four and sealed envelopes prepared by an independent statistician (Figure 1). Socio-demographic variables, such as age, marital status, and living arrangements, were considered. The following clinical characteristics were evaluated: cognitive status by a 6-item orientation–memory–concentration test, the Short Blessed Test (SBT) [34]; the physical burden of illness, in particular, the severity and comorbidity index (clinical and functional severity of 14 categories of diseases) assessed by the Cumulative Illness Rating Scale CIRS-s and CIRS-c, respectively [35]; and glomerular filtration rate (using the chronic kidney disease epidemiology collaboration formula) [36]. Data from the disease activity score (DAS 28) [37], health assessment questionnaire (HAQ) [38,39], and short-form health survey (SF-12) “Italian version” were collected and used as outcome variables [40,41,42,43]. All the outcome measures were evaluated by a physician regarding interventions.

### 2.3. Intervention

#### Occupational Therapy and Exercise Elements

The elements of OT of this study concerned: the correct positioning of the hands toward objects, better management of the pauses between the executions of daily activities, the better balance of the body and sections when carrying heavy objects, the management of efforts, and better use of tools for the development of daily activities.

The protocol was divided into two parts: (1) physical exercise to manage balance to maintain muscle tone and joint restart after the night, and (2) joint protection and decreased joint load during daily actions. The physical exercise included the flexion and extension of the upper and lower limbs for the main joints (ankle, knee, cox-femoris, elbow, wrist, and shoulder) with or without weight tools for 20/30 min day (Table 1). Joint protection included the positioning of the body correctly aligned to reduce the load, using facilitators for simple tools for people with deformed fingers, and tips for handling during cooking, dressing, walking, washing, and shopping (Table 1).
-General Health (GH)

GH represents the evaluation of the patient’s global health status, with a scale of 0 to 100, where 0 indicates the best, and 100 represents the worst state of health.
-Disease Activity Score (DAS 28)

DAS 28 is an index of RA activity evaluation based on the measurement of synovitis present in 28 joints. A score of (>5.1) determines a high disease activity, between (3.2–5.1) moderate activity, (2.6–3.2) low activity, and <2.6 indicates clinical remission. Furthermore, the DAS 28 result was performed using both the C-reactive protein (CRP) (DAS 28crp) and erythrocyte sedimentation rate (ESR) (DAS 28esr).
-Health Assessment Questionnaire (HAQ)

HAQ is used for the determination of physical disability. This questionnaire includes 20 questions describing the ability to perform daily actions during the last week, such as dressing and grooming, arising, eating, walking, hygiene, reaching, gripping, errands, and chores. This score allows us to divide the degree of physical disability into categories: (0–0.49) no disability, (0.5–0.99) mild disability, (1.0–1.99) moderate disability, and 2 < severe disability.
-Questionnaire on physical health Italian version (SF-12)

The SF-12 consists of 12 items from the larger SF-36, and it is used to measure health status in the general population. The SF-12 measures physical functioning, role limitations due to physical health problems, bodily pain, general health, vitality (energy/fatigue), social functioning, role limitations due to emotional problems, and mental health (psychological distress and psychological well-being). Two composite scores—the physical component score (PCS) and the mental component score (MCS)—are computed from all 12 items using a standard scoring algorithm. The PCS score primarily focuses on physical functioning, physical roles, bodily pain, and general health, and vitality scales. The MCS focuses on the scales of vitality, social functioning, emotional roles, and emotional well-being. PCS and MCS scores range from 0 to 100; a higher score indicates a better health status.

Furthermore, an information booklet was created to be provided to patients, with the elements of the joint economy protocol to be performed.

During the first day, a brochure was provided to the patients in the intervention group, and a nurse and a kinesiologist explained the joint economy and how to perform the gestures. The brochure contains brief information in the Italian language about rheumatoid arthritis and its comorbidities, the therapies available with side effects, indications for different treatments, and an in-depth explanation of the exercises to perform at home for six days a week, with one day of rest per week, the time to devote (15 min) to each exercise, and the number of repetitions to perform (up to ten).

On the first meeting, only the brochure was delivered to the control group patients, advising them to read it carefully. All patients (IG and CG) were evaluated after 30 days.

### 2.4. Statistical Analysis

Data were reported as percentages for categorical variables and as means (95% confidence intervals) for quantitative variables. The comparison between groups was conducted using Fisher’s exact test for contingency tables and the z-test for the comparison of proportions. A non-parametric Mann–Whitney U test was used to compare quantitative variables between groups. The Wilcoxon signed-ranked test was used for post- versus baseline comparisons.

A two-tailed *p* < 0.05 was considered statistically significant. Stata Statistical Software: Release 14.1, College Station, TX, USA: StataCorp LP) was used for database management and analysis. The data analysis was conducted by a statistician blinded to the intervention and control groups.

## 3. Results

A total of 160 patients with a diagnosis of rheumatoid arthritis were eligible for this analysis; among them, 86 were women, and 74 were male. The mean age was 58.0 (42.4–74.7) for the control group and 54.0 (39.7–68.3) for the intervention group. Table 2 shows the two study groups’ demographic characteristics and modifiable risk factors. Almost half of the patients in the two groups were married, and one-third of the control and intervention groups lived alone. Patients in the intervention group had a higher cumulative illness rating scale when evaluating the severity and comorbidity index, even if not significant, as well as morning stiffness. Before randomization, DAS 28 calculated both the C-reactive protein and erythrocyte sedimentation rate; HAQ and both SF-12MCS and SF-12PCS were not significant. Overall, disease distribution showed that arterial hypertension, fibromyalgia, diabetes mellitus, COPD, and chronic kidney disease were more frequent. After 30 days of intervention, DAS 28 esr, DAS 28 crp, HAQ, and SF-12 MCS (*p* = 0.005, 0.004, 0.009, and 0.010, respectively) were found to be statistically significant, while SF-12 PCS improved even if no significant difference was found (Table 3). The intervention had statistically significant effects (*p* < 0.001) after the comparative analysis of the mean differences between the baseline and the post-intervention values among the control and intervention groups (Table 4).

## 4. Discussion

This study aimed to evaluate the impact of physical exercise and an OT program on the quality of life in patients with RA. Our results indicate that a correct one-month educational intervention at home can improve the quality of life of patients with RA.

RA is a systemic disease characterized by inflammation and the involvement of synovial joints of the musculoskeletal system [44]. This disease has significant consequences for people, causing a loss of function and working disability, and poses a substantial healthcare burden on both the individual and society [45,46].

Although the availability of traditional and biologic disease-modifying antirheumatic therapies (DMARDs) can decrease this burden significantly [47], remission may not be reasonably achievable in some patients with RA, especially in subjects with established disease [48,49,50].

Hypotrophy and muscle hyposthenia are frequent and often related to the direct involvement of disease processes and steroid therapy [47]. At the same time, the analgesic attitudes put in place by patients to avoid pain can cause secondary damage due to the postural and gestural changes developed to compensate for the initial disability, which can sometimes be more severe than the initial damage. A vicious circle can therefore be established that leads to the worsening of physical activity and deconditioning.

To interrupt this vicious circle, the optimal treatment of rheumatoid arthritis involves a multidisciplinary approach, including physical exercise therapy, occupational therapy, patient self-management, and pharmacological therapy. The fitness and physical exercise of patients with RA are significantly lower than those without RA. However, the benefits of physical activity and exercise substantially improve various aspects of physiological mechanisms and the quality of life in RA and other diseases [51], decreasing the evolution of overall joint involvement, treating the musculoskeletal system as a whole, and maintaining adequate fitness [52].

Physical exercise programs include improved cardiorespiratory fitness and cardiovascular health, increased muscle mass, reduced adiposity (including attenuated trunk fat), improved strength, and physical functioning, and are all achieved without exacerbating disease activity or joint damage.

Different studies have shown that any exercise is better than no exercise, but the intensity, frequency, and movement period for better results are not determined [52,53,54].

The American College of Rheumatology guidelines [55] recommend exercise, which includes a range of motion exercises and resistance training to preserve joint mobility and maintain muscle mass, bone health, and fitness. In particular, resistance training is related to anabolic effects in muscles [56]. In addition, the local production of IGF-1 [57] and growth factors in response to muscle contraction play a crucial role because they are potent activators of this pathway. The anti-inflammatory effects of exercise may also contribute to decreased ESR and the beneficial effects observed on joints.

Our data agree with previous studies showing that aerobic cardiorespiratory conditioning improved HAQ with better results in the case of established RA and short-term programs [52], as opposed to randomized controlled trials that showed no significant differences between the groups for the HAQ DI score in older adults [58]. Our results showing the benefit of aerobic exercises in middle-aged RA subjects should not be generalized to older RA patients. However, other studies showed that an individually adapted training session with moderate-to-high intensity exercise with person-centered guidance decreased fatigue, improved symptoms of depression, and was accompanied by metabolic changes in older adults with RA [59]. Our results highlight that home-based exercises could facilitate quality of life and disease-related parameters in contrast with previous interventions [60]. Several lines of the literature showed that supervised physical exercises resulted in a higher adherence to the program with a better trend in the quality of life, disease activity, and pain score [61].

However, the impact of supervised dynamic exercise programs on work and the consumption of medical and paramedical resources is still being determined. Van den Hout et al. suggested that supervised class exercises [62] were more expensive than home-based interventions [63] and provided insufficient improvement to justify the additional costs. Our results align with Manning and colleagues, which showed that education, self-management, and upper limb exercise training represent a cost-effective use of resources compared with usual care and lead to lower healthcare costs and work absence in patients with RA [64]. In addition, an individually tailored intervention aimed at reducing sedentary behavior in patients with RA improved participants’ 22-month health status and reduced healthcare costs [65].

The physical activity coaching intervention resulted in an improved effect on the visual analog scale score (VAS) for the intervention group at a higher cost. To maximize cost-effectiveness, this type of physical activity coaching intervention should be targeted toward patients primarily affected by their RA.

Our study supports the utilization of occupational therapy. Previous data showed that joint protection and energy conservation improved functionality, pain, and work satisfaction in comparison with subjects who did not receive the interventions [66].

This approach was first described in 1965 [67] through the analysis of motor impairments motivated by the inflammatory process typical to RA and its combination with biomechanical principles, aiming to minimize the action of forces that favored the development of joint deviations and deformities while performing daily tasks [68], for example, the hyperextension of the metacarpophalangeal joint of the finger I, ulnar deviation of metacarpophalangeal joints of the fingers II–V, and installation of deformity standards, such as swan neck, hammer toe, or buttonhole toe, through the involvement of distal interphalangeal joints [69].

OT can be carried out with exercise programs, joint protection, counseling, and device instruction [70]. OT interventions for rheumatoid arthritis patients (classified as comprehensive therapy, training of motor function, training of skills, instruction on joint protection and energy conservation, counseling, education about assistive devices, and provision of splints) improve outcomes on functional ability, social participation, and health-related quality of life.

Randomized trials with high levels of evidence [71,72] on the effectiveness of joint protection and energy conservation methods showed a significant improvement in pain reduction among patients receiving guidelines for changes in their activities of daily living (ADL) [26,28,31]. An improvement in fatigue and increased social participation [31], a reduction in morning stiffness, a lower incidence of deformities in the hands [30], and improved functionality were observed, even among patients with a severe RA state [27].

The early treatment conducted by a multidisciplinary team is an effective method to minimize complications related to work, maintaining the working capacity of these patients for a while, which is similar to that found among the healthy population [73,74,75,76,77]: the correct positioning of the hands toward objects, the better management of pauses between the executions of daily activities, the better balance of the body and sections when carrying heavy objects, the control of efforts, and better use of tools to develop daily activities.

A recent systematic review highlighted the focus of ‘renegotiating the self’ as a central concern in RA self-management, suggesting standardized self-management programs focused on disease management and daily functioning [78].

In this sense, different studies have shown that quality of life can be related to the perception of the components of psychological and physical health concerning one’s level of self-sufficiency and environmental–social relations [79,80,81]. The information on these parameters is essential for the documentation and evaluation of the therapies and the health status of patients.

In this study, the parameters analyzed to evaluate disease activity and health showed significant variation compared to the start of the study. In particular, in IG, HAQ, DAS 28 esr, and DAS 28 crp, SF-12 MCS significantly improved compared with CG, while SF-12 PCS ameliorated, even if not significantly compared to pre-treatment. Our results are in agreement with those of previous studies [19,82,83] that showed how exercise and occupational therapy in a patient with RA could improve their state of health, including, in particular [75], scales measuring aspects of mental functioning and physical aspects or mobility, role limitation due to physical health problems and usual activities, and bodily pain. In addition, this study implemented knowledge because it shows how the correct execution of the exercises, after explanation, amplifies the benefits of these activities.

In this study, some limitations have to be outlined. First of all, this is a single-center analysis. The population study consisted only of patients recruited in our rheumatology outpatient clinic. On the contrary, the major strength of our research lies in the rigorous data collection in a standardized fashion. Although the sample size was limited, the results are quite robust to support further research. Our observations need to be verified by later studies to avoid population bias due to it being a single-center study.

## 5. Conclusions

Our results reveal that a one-month self-management program of joint education, including physical exercise and OT at home, showed how patient education is an integral part of the disease management process. In particular, self-management causes the patient to increase self-efficacy, which is understood as an improvement in independence and quality of life. Moreover, the correct execution of physical exercises plays a fundamental role in managing RA symptoms. For this reason, it is important that a professional figure, such as an occupational therapist and kinesiologist, is present in the clinical department. Further research needs to confirm and generalize our findings.

## Figures and Tables

**Figure 1 healthcare-11-02123-f001:**
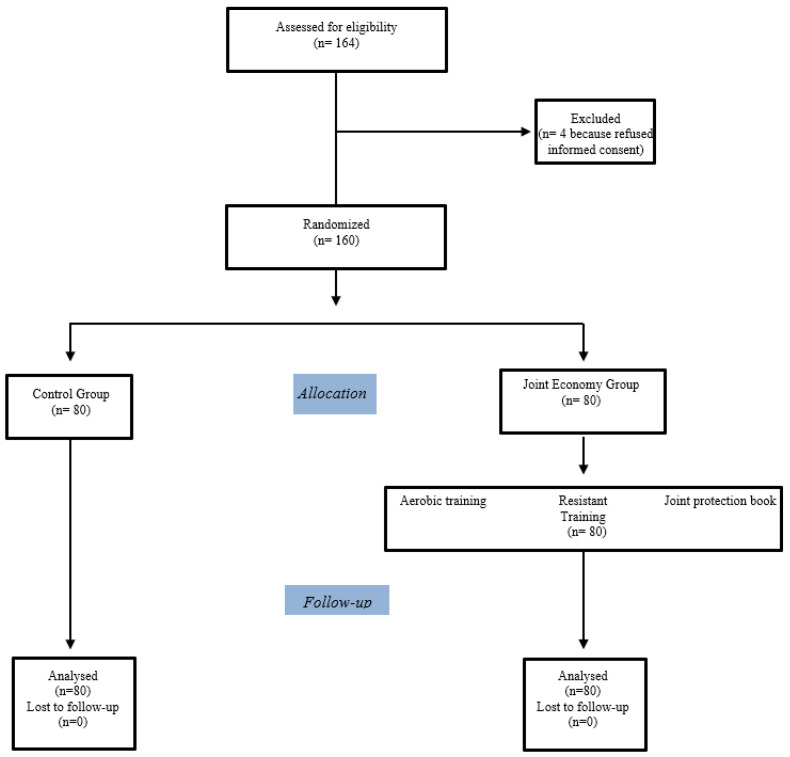
Flow diagram according to the Consort Statement of progress through the phases of the study groups (enrolment, intervention allocation, follow-up, and data analysis). The number of patients (n) screened for eligibility, excluded, randomized, and allocated to the study groups is reported.

**Table 1 healthcare-11-02123-t001:** Description of the exercises performed for the joint economy method.

Physical Exercise	
Joint Movement *	Muscle Tone **
Extention ankle	Biceps
Flextion ankle	Shoulder
Exsention knee	Deltoids
Flextion knee	Legs
Exsention coxo-femoris	Quadriceps
Flextion cvoxo-femoris	Handle grip
Trunck rotation (right–left)	
Exsention elbow	Aerobic exercise ***
Flextion elbow	Walk
Exsention wrist	Cycle ergometer
Flextion wrist	
Exsention shoulder	
Flextion shoulder	

* Joint movement: 2 sets and 10 repetitions. ** Muscle tone: 2 sets and 8–10 repetitions at 70% of 1 RM. *** Aerobic exercise: 10’ walking and/or cycle ergometer. Each exercise was performed considering the level of pain.

**Table 2 healthcare-11-02123-t002:** Basic socio-demographic variables, clinical characteristics, disease distribution and general health (GH) evaluation, disease activity score (DAS 28), health assessment questionnaire (HAQ), and short-form health survey (SF-12) “Italian version” among patients with rheumatoid arthritis (RA) that were administered the joint economy intervention (intervention group) and patients with RA that were not administered the joint economy intervention (control Group).

Variable	Control Group	Intervention Group	*p*
N	80	80	-
Age (Y)	58.0 (42.4–74.7)	54.0 (39.7–68.3)	0.121
BMI	31.3 (30.2–32.4)	28.4 (21.5–35.3)	0.243
Years of illness (Y)	12.1 (0.1–18.3)	18.0 (7.9–28.2)	0.071
Marital status (%)			0.713
Married	52.2	51.3	
Separeted	23.6	24.5	
Divorced	20.6	20.6	
Widow	3.4	3.6	
Living arrangement			0.443
Alone	29.8	30.7	
Spouse	26.3	25.2	
Spouse and sons	27.4	27.2	
Sons	15.5	15.9	
Other	1.0	1.0	
Morning stiffness	25.0 (8.1–42.3)	33.8 (12.3–54.6)	0.178
DAS 28 esr	5.7 (4.7–6.7)	5.3 (4.7–5.9)	0.301
DAS 28 pcr	5.6 (4.9–6.3)	5.2 (4.4–6.0)	0.356
HAQ	1.6 (0.5–2.7)	1.9 (1.3–2.5)	0.265
ESR	17.3 (0.3–34.6)	18.0 (3.9–12.1)	0.060
CRP	5.2 (0.8–9.6)	5.7 (0.4–3.0)	0.058
SF 12 PCS	26.2 (22.4–30.0)	28.1 (24.8–31.7)	0.654
SF 12 MCS	42.9 (32.4–53.4)	42.8 (37.9–47.7)	0.834
Hypertension (%)	56.1	56.9	0.901
Fibromyalgia (%)	35.3	38.2	0.345
Diabetes (%)	31.3	30.0	0.532
COPD (%)	24.7	25.4	0.865
Kidney Disease (%)	20.9	20.2	0.893
Depression (%)	12.6	14.1	0.643
Heart Failure (%)	11.2	10.9	0.763
Ischemic Heart Disease (%)	11.0	10.8	0.872
Obesity (%)	7.3	7.5	0.905
Anemia (%)	7.2	6.8	0.785
Liver Disease (%)	5.3	5.5	0.891
Atrial Fibrillation (%)	4.1	3.9	0.910
Osteoarthritis (%)	3.3	3.4	0.956
Severity Index (by CIRS)	2.58	2.81	0.1223
Comorbidity index (by CIRS)	2.59	2.91	0.3510
GFR (mL/min)	59.6 (58.4–60.3)	58.8 (57.3–60.6)	0.3902
SBT	2.7 (1.0–3.3)	1.4 (0.8.0–3.1)	0.1049

Data are reported as the mean (95% confidence interval). The non-parametric Mann–Whitney U test was used for comparisons. No variable comparison resulted in statistical significance. Disease activity score esr (DAS 28 esr), disease activity score pcr (DAS 28 pcr), Health Assessment Questionnaire (HAQ), Short-Form Health Survey-12 physical component score (SF-12 PCS), Short-Form health survey-12 mental component score (SF-12 MCS), Erythrocyte Sedimentation Rate (ESR), C-reactive protein (CPR), Cumulative Illness rating scale (CIRS), Glomerular filtration rate (GFR), Short Blessed Test (SBT).

**Table 3 healthcare-11-02123-t003:** Indexes of disease activity and quality of life in patients with Rheumatoid Arthritis according to the administration of joint economy intervention.

**Control Group**			
**Variable**	**Baseline**	**Post-** **Intervention**	** *p* **
DAS28 esr	5.7 (4.7–6.7)	5.6 (5.0–6.2)	*p* = 0.610
DAS28 crp	5.6 (4.9–6.3)	5.4 (4.8–6.0)	*p* = 0.574
HAQ	1.6 (0.5–2.7)	1.5 (0.6–2.4)	*p* = 0.600
SF-12 MCS	42.9 (32.4–53.4)	43.8 (39.6–48.0)	*p* = 0.683
SF-12 PCS	26.2 (22.4–30.0)	26.6 (22.1–30.2)	*p* = 0.414
**Intervention Group**			
**Variable**	**Baseline**	**Post-Intervention**	** *p* **
DAS28 esr	5.3 (4.7–5.9)	4.8 (3.9–5.7)	*p* = 0.005
DAS28 crp	5.2 (4.4–6.0)	4.6 (3.3–5.9)	*p* = 0.004
HAQ	1.9 (1.3–2.5)	1.6 (1.0–2.2)	*p* = 0.009
SF-12 MCS	42.8 (37.9–47.7)	48.0 (43.8–52.2)	*p* = 0.010
SF-12 PCS	28.1 (24.8–31.7)	34.9 (30.5–40.1)	*p* = 0.093

The Wilcoxon signed-ranked test was used for post-intervention versus baseline comparisons. The bold indicates a statistically significant result.

**Table 4 healthcare-11-02123-t004:** Mean differences between post-intervention and baseline values used to establish the efficacy of the joint economy intervention.

	Control Group	Intervention Group	
Variable	Mean Difference ± SD	Mean Difference ± SD	*p* *
DAS28 esr	0.375 ± 1.69	1.95 ± 0.84	**<0.** **001**
DAS28 crp	0.70 ± 1.69	2.0125 ± 0.88	**<0.** **001**
HAQ	0.35 ± 1.71	2.05 ± 0.83	**<0.** **001**
SF-12 MCS	1.41 ± 2.94	3.575 ± 1.10	**<0.** **001**
SF-12 PCS	0.61 ± 2.89	3.4375 ± 1.18	**<0.** **001**

The Wilcoxon signed-ranked test was used for post-intervention versus baseline comparisons.* *p* values were computed by the Mann–Whitney U test for comparisons between the control and intervention groups. The bold indicates a statistically significant result.

## Data Availability

Data are unavailable due to privacy and ethical restrictions.

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
