# Peer review of "Physical Exercise and Occupational Therapy at Home to Improve the Quality of Life in Subjects Affected by Rheumatoid Arthritis: A Randomized Controlled Trial"

_healthcare, 2023, doi:10.3390/healthcare11152123_

Round 1
Reviewer 1 Report
1. The reason for choosing one month as the intervention time is not clear, so it is suggested to increase the reason why we choose one month as the intervention time instead of two months or even longer;
2. The introduction of exercise interventions is vague. So it is suggested to point out what kind of physical exercise and occupational therapies should be selected, how long the time of each exercise or treatment should be and so on;
3. The data processing method was fuzzy, and the abstract did not specify which statistical method was selected to obtain the difference between the score scales of the control group and the intervention group, so it is suggested to add a clear data statistical method;
4. The general health (GH) evaluation was selected in the paper, but the results of GH evaluation data were not mentioned, so it is suggested to increase the results related to GH evaluation.
5. Some data mentioned in the discussion were not presented in the results.For example, the data in this paper are consistent with previous studies that aerobic cardiopulmonary conditioning can improve HAQ, with better results in established RA and short-term treatment regiments, but the data are not presented in the results.It is recommended to add relevant data to the results.
The English quality of this paper is good, readers can quickly understand what the author wants to express after reading, but there are still some individual sentences are too long, too many modified words , which is difficult to understand the meaning.So a little modification of English expressions is needed.
Author Response
REVIEWER 1
Thank you for your comments helpful to improve the quality of our manuscript.
1. The reason for choosing one month as the intervention time is not clear, so it is suggested to increase the reason why we choose one month as the intervention time instead of two months or even longer;
Thank you for your observation. This is a pilot study which should be extended to assess short-, medium- and long-term effects. As pilot study it wants to verify the potential positive results at one month.
2. The introduction of exercise interventions is vague. So, it is suggested to point out what kind of physical exercise and occupational therapies should be selected, how long the time of each exercise or treatment should be and so on;
Thank you for the punctual observation that allows us to be clearer. In table 1 there is a detailed description of the exercises performed and the number of sets and repetitions that patients must perform at home daily, for six days a week, with one day of rest per week. We reported in the text.
3. The data processing method was fuzzy, and the abstract did not specify which statistical method was selected to obtain the difference between the score scales of the control group and the intervention group, so it is suggested to add a clear data statistical method;
Thank you for your observation. We added the use of wilcoxon in the statistics section. We made this correction because we realized that this part is skipped. Moreover, according to your suggestion we specified statistical methods in the abstract.
4. The general health (GH) evaluation was selected in the paper, but the results of GH evaluation data were not mentioned, so it is suggested to increase the results related to GH evaluation.
Thank you for your observation. We apologize for the error but the general health data are not available for all patients. For this reason, we have provided to remove it from the text.
5. Some data mentioned in the discussion were not presented in the results. For example, the data in this paper are consistent with previous studies that aerobic cardiopulmonary conditioning can improve HAQ, with better results in established RA and short-term treatment regiments, but the data are not presented in the results. It is recommended to add relevant data to the results.
Thank you for your observation. Data about HAQ are reported in the results section at the end of the paragraph. We do not have specific data on the aerobic cardiopulmonary conditioning. In this sense we modified the discussion section

Reviewer 2 Report
Dear authors,
Thank you for submitting your work. Please find my comments, make necessary edits as per reviewer(s) comments, and resubmit.
The title does not reveal what type of article this is. As it is an RCT, mention that.
In the abstract, the methods does not mention what intervention is being provided to IG. This is important because the first thing the readers go through is an abstract.
The introduction is very lengthy, with 29 references. Please establish a premise, mention the hypothesis, and move the detailed introduction part to the discussion with the references.
Outcomes measures were evaluated by a physician blinded regarding interventions- this is mentioned in the methods. What about the physicians who analyzed the data? Where they blinded? If yes, mention this. If no, mention this in the limitations.
How was the normal/skewed distribution of the data determined? Please mention in the sub-heading: statistical analysis.
The results section should begin with the mention of CONSORT flow diagram with the citation to the image/figure/document.
After the table mention the statistical tests that were used for the various outcomes/variables as a footnote and use symbols to highlight in the p-value section.
There are many small paragraphs, with 2-3 sentences only in the discussion. Please organize the areas with similar issues.
The study was not prospectively registered with any clinical trials registry. This is another limitation.
Overall, the quality of English is fine. However, if accepted, might need technical edits, to comply to general instructions.
Author Response
REVIEWER 2
Thank you for your comments helpful to improve the quality of our manuscript.
The title does not reveal what type of article this is. As it is an RCT, mention that.
Thank you for your observation. According to your suggestion we added the type of study in the title.
In the abstract, the methods does not mention what intervention is being provided to IG. This is important because the first thing the readers go through is an abstract.
Thank you for your punctual observation. According to your suggestion we mentioned the intervention in the abstract.
The introduction is very lengthy, with 29 references. Please establish a premise, mention the hypothesis, and move the detailed introduction part to the discussion with the references.
Thank you for your punctual observation. According to your suggestion we modified the introduction section and the discussion section.
Outcomes measures were evaluated by a physician blinded regarding interventions- this is mentioned in the methods. What about the physicians who analysed the data? Where they blinded? If yes, mention this. If no, mention this in the limitations.
Thank you for your punctual observation. According to your suggestion we reported in the text that data analysis was made by a statistician blinded regarding the intervention and control groups.
How was the normal/skewed distribution of the data determined? Please mention in the sub-heading: statistical analysis.
Thank you for your observation. No evaluation of data distribution was made because the sample size is small and false negative data would be too frequent, for this reason non-parametric or distribution free tests were used.
The results section should begin with the mention of CONSORT flow diagram with the citation to the image/figure/document.
Thank you for your observation. The CONSORT flow diagram was inserted at the beginning of material and methods section.
After the table mention the statistical tests that were used for the various outcomes/variables as a footnote and use symbols to highlight in the p-value section.
Thank you for your observation. According to your suggestion we mentioned the statistical tests used.
There are many small paragraphs, with 2-3 sentences only in the discussion. Please organize the areas with similar issues.
Thank you for your observation. According to your suggestion we reorganize the discussion section
The study was not prospectively registered with any clinical trials registry. This is another limitation.
Thank you for your observation. As reported in the section material and methods The present study was developed in accordance with the Declaration of Helsinki and had approval from the Ethics committee of Palermo 2 with the approval number 18-2021.

Reviewer 3 Report
Manuscript Number:
Physical Exercise and Occupational Therapy at Home to Improve the Quality of Life in Subjects Affect by Rheumatoid Arthritis
Reviewer :
This study investigated the role of a 30-days joint economy intervention (integration of physical exercise and occupational therapy) at home on the quality of life of subjects with Rheumatoid Arthritis (RA). The reporting of the study followed the guidelines of the Consolidated Reporting of Trials (CONSORT). I think the data are interesting. Article appropriate to Healthcare Journal. Research question was relevant. Cited literature pertinent. Appropriate data analysis. Discussion clear with new insights. Recommendation: Accept with justification of clinical relevance of this manuscript and include the manuscript hypothesis.
Thank you for giving me the opportunity to review this manuscript. This manuscript is quite interesting and present new data. I found the paper interesting to read and congratulate the authors for this effort.
Before publication, some issues should be addressed:
Introduction:
- Please, to improve the background and rationale, mention previous studies.
- Physical exercise programs. Explain. Include figures.
- Include the manuscript hypothesis.
- The clinical relevance and the rationale for the study need to be strengthened. What's new in the scientific literature with this manuscript? Improve.
Methods:
- Please, describe a bit more some characteristics from population. Participant selection.
- One hundred and sixty outpatients with rheumatoid arthritis attending the Rheumatologic ambulatory were enrolled (86 females and 74 men). The selection criteria included a diagnosis of Rheumatoid Arthritis, age ≥ 18 years 103 old subjects. Why used this classification? Diagnostic assessment? Describe.
- The optimal treatment of rheumatoid arthritis involves a multidisciplinary approach, including physical exercise therapy and occupational therapy. Who performed the exercises? The physiotherapist or the occupational therapist? Professional experience time?
- And the comparison between the control group and the intervention group? It is important to support the use of the instrument and clearly and objectively justify the reason for performing this type of analysis.
- Discussion
- Include the implications/applications of the study for the field movement sciences.
Moderate editing of English language required.
Author Response
REVIEWER 3:
Thank you for your comments helpful to improve the quality of our manuscript.
Introduction:
• Please, to improve the background and rationale, mention previous studies.
• Physical exercise programs. Explain. Include figures.
• Include the manuscript hypothesis.
• The clinical relevance and the rationale for the study need to be strengthened. What's new in the scientific literature with this manuscript? Improve.
Thank you for your observation. According to your suggestion we improve the introduction section
Methods:
• Please, describe a bit more some characteristics from population. Participant selection.
• One hundred and sixty outpatients with rheumatoid arthritis attending the Rheumatologic ambulatory were enrolled (86 females and 74 men). The selection criteria included a diagnosis of Rheumatoid Arthritis, age ≥ 18 years 103 old subjects. Why used this classification? Diagnostic assessment? Describe.
Thank you for your observation. We included subjects with eighteen years or older because we do not take care of children or teenagers.
• The optimal treatment of rheumatoid arthritis involves a multidisciplinary approach, including physical exercise therapy and occupational therapy. Who performed the exercises? The physiotherapist or the occupational therapist? Professional experience time?
Thank you for your observation. In our department, an expert kinesiologist performed exercises.
• And the comparison between the control group and the intervention group? It is important to support the use of the instrument and clearly and objectively justify the reason for performing this type of analysis.
Thank you for your observation. According to your suggestion the comparison between groups was made and reported in table 2. The intragroup variables are those that determine the results; therefore, we do not agree on comparing the post intervention between the two groups, however we have specified the type of text used.
• Discussion
• Include the implications/applications of the study for the field movement sciences.
Thank you for your observation. According to your suggestion we included implications in the conclusion section.

Reviewer 4 Report
healthcare-2448883_review
Title: Physical Exercise and Occupational Therapy at Home to Improve the Quality of Life in Subjects Affect by Rheumatoid Arthritis
Comments for Authors
Dear authors,
I have carefully read your paper, which investigated the effects of a home intervention of joint economy with physical exercise and occupational therapy, on the quality of life of subjects with rheumatoid arthritis applied over a period of one month.
In your results an improvement in the disease activity score, health assessment questionnaire and the mental component in the short-form health survey in the intervention group after the intervention. Therefore, it appears that an intervention combining physical exercise and occupational therapy positively affects patients' quality of life with rheumatoid arthritis. In my opinion, more studies are needed from the point of view of the occupational therapy with interventions of this type to promote joint economy in these patients, with a larger sample size and a longer follow-up time to support these results.
In general, the text is understandable. However, I found some issues in the methodology, discussion and results sections that should be addressed to improve the paper, in my opinion.
Specific comments:
Abstract
- Page 1, line 32. I suggest you add the full term of ers and crp.
- Page 1, line 33-34. I suggest you remove the phrase " in comparison with people received the usual care" as the results do not show these data on the between-group analysis.
Introduction
- Page 1-2. I suggest you add more information about the pathology, age ranges in which it is more frequent, prevalence according to sex, etc.
- Page 2, lines 60-62. Please, could you add more information about the definition of occupational therapy at this point? Although in recent years, great advances have been made in this field, however I believe that it is still necessary to give it more visibility.
- Page 2, line 76. Please, add the necessary references that support the information shown in this paragraph
- Page 2, lines 82-83. You have specified the hypothesis of your study. However, I suggest that you reformulate it in a more precise way.
Methods
- How was sample size determined? Please, add this information, is possible.
- Page 2, line 96. Please, could you add information about the period of time and places in which the data were recorded? Where were the participants evaluated? Did the participants complete the questionnaires at home, at the university, hospital...? How long did it take to complete the entire assessment?
- Pages 3-4, lines 103-164. The information that appears in the methodology section is disordered; it makes difficult read the manuscript. I suggest you rearrange it. You could add a title to introduce the analysed variables. First explain the variables and later the intervention with the table 1. Please, also add a brief description of the Short-Blessed-Test and the Cumulative-Illness-Rating-Scale CIRS-s and CIRS-c.
- Page 4, line 159. You mentioned that an information booklet was created, please add more information or picture about it.
Page 4, lines 161-162. It has caught my attention that the person who explains the joint economy to patients is a nurse. Why didn't an occupational therapist perform this intervention? It would be the most indicated since you speak of an occupational therapy hospitalization.
It is not clear how you carried out the intervention, I have many doubts about it, who was in charge of performing the intervention? and where it was applied? please add more information to clarify these points. Did the patients perform the exercises every day for 30 days? during how much time? The patients only received an information session together with the booklet and then they did the exercises at home? How could you control that the patients performed the exercises properly? I understand that to make the necessary adaptations an occupational therapist should go to the homes of the patients…
Results
- Page 5, line 186. In these lines you mentioned “even if not significantly, as well as morning stiffness”, please add the p value of this variables in the text.
- You mention in your statistical analysis that you made the comparison between the groups; however, this information does not appear in the text of the results. Table 3 shows us the results in the control group and table 4 in the experimental group, separately, that is, the intragroup results. However, no information or table appears about the comparison or difference of the means for these variables between the groups. I believe that this information is crucial for your manuscript and could enrich it considerably.
- Discussion
- Page 8, line 263, line 280. Although VAS and ADLs are well known terms, to make the manuscript easier to read for people who may not be familiar with them add the full terms.
- Page 8, line 287. Grammar mistake, capital letter after period. Please check it.
- Page 8, line 301. As I have mentioned before, you do not provide results of the comparison between the groups, therefore you cannot affirm that the improvement that occurs in this parameter is significant or not.
Conclusions
- Page 11. I suggest you to the reformulate your conclusions in a more carefully way such as: “Our results suggest that ….”
I hope that my comments could help to improve the paper.
Minor editing of English language required
Author Response
REVIEWER 4
Thank you for your comments helpful to improve the quality of our manuscript.
Abstract
- Page 1, line 32. I suggest you add the full term of ers and crp.
- Page 1, line 33-34. I suggest you remove the phrase " in comparison with people received the usual care" as the results do not show these data on the between-group analysis.
Thank you for your observation. According to your suggestion we modified the text.
Introduction
- Page 1-2. I suggest you add more information about the pathology, age ranges in which it is more frequent, prevalence according to sex, etc.
- Page 2, lines 60-62. Please, could you add more information about the definition of occupational therapy at this point? Although in recent years, great advances have been made in this field, however I believe that it is still necessary to give it more visibility.
- Page 2, line 76. Please, add the necessary references that support the information shown in this paragraph
- Page 2, lines 82-83. You have specified the hypothesis of your study. However, I suggest that you reformulate it in a more precise way.
Thank you for your observation. According to your suggestion we modified the text.
Methods
- How was sample size determined? Please, add this information, is possible.
Thank you for your observation. This is a pilot study which should be extended to assess short-, medium- and long-term effects. As pilot study it wants to verify the potential positive results at one month.
- Page 2, line 96. Please, could you add information about the period of time and places in which the data were recorded? Where were the participants evaluated? Did the participants complete the questionnaires at home, at the university, hospital...? How long did it take to complete the entire assessment?
Thank you for your observation. Data were recorded and participants were evaluated at the Internal Medicine Department of ARNAS Civico-Di Cristina-Benfratelli Hospital of Palermo in the outpatient service.
- Pages 3-4, lines 103-164. The information that appears in the methodology section is disordered; it makes difficult read the manuscript. I suggest you rearrange it. You could add a title to introduce the analysed variables. First explain the variables and later the intervention with the table 1. Please, also add a brief description of the Short-Blessed-Test and the Cumulative-Illness-Rating-Scale CIRS-s and CIRS-c.
Thank you for your observation. According to your suggestion we modified the text.
- Page 4, line 159. You mentioned that an information booklet was created, please add more information or picture about it.
Thank you for your observation. According to your suggestion we added information in thetext.
Page 4, lines 161-162. It has caught my attention that the person who explains the joint economy to patients is a nurse. Why didn't an occupational therapist perform this intervention? It would be the most indicated since you speak of an occupational therapy hospitalization.
Thank you for your observation. In our department a kinesiologist along with a nurse explained the joint economy to patients.
It is not clear how you carried out the intervention, I have many doubts about it, who was in charge of performing the intervention? and where it was applied? please add more information to clarify these points. Did the patients perform the exercises every day for 30 days? during how much time? The patients only received an information session together with the booklet and then they did the exercises at home? How could you control that the patients performed the exercises properly? I understand that to make the necessary adaptations an occupational therapist should go to the homes of the patients…
Thank you for your observation. According to your suggestion we provided more information to clarify your doubts
Results
- Page 5, line 186. In these lines you mentioned “even if not significantly, as well as morning stiffness”, please add the p value of this variables in the text.
- You mention in your statistical analysis that you made the comparison between the groups; however, this information does not appear in the text of the results. Table 3 shows us the results in the control group and table 4 in the experimental group, separately, that is, the intragroup results. However, no information or table appears about the comparison or difference of the means for these variables between the groups. I believe that this information is crucial for your manuscript and could enrich it considerably.
Thank you for your observation. According to your suggestion the comparison between groups was made and reported in table 2. The intragroup variables are those that determine the results, therefore we do not agree on comparing the post intervention between the two groups, however we have specified the type of text used.
- Discussion
- Page 8, line 263, line 280. Although VAS and ADLs are well known terms, to make the manuscript easier to read for people who may not be familiar with them add the full terms.
- Page 8, line 287. Grammar mistake, capital letter after period. Please check it.
- Page 8, line 301. As I have mentioned before, you do not provide results of the comparison between the groups, therefore you cannot affirm that the improvement that occurs in this parameter is significant or not.
Thank you for your observation. According to your suggestion we modified the text.
Conclusions
- Page 11. I suggest you to the reformulate your conclusions in a more carefully way such as: “Our results suggest that ….”
I hope that my comments could help to improve the paper.
Thank you for your observation. According to your suggestion we modified the text.

Round 2
Reviewer 2 Report
Dear authors,
Thank you for revising the manuscript based on the comments raised.
The English language needs minor technical edits.
Author Response
Thank you for revising the manuscript based on the comments raised.
Once again, thank you for your comments helpful to improve the quality of our manuscript

Reviewer 4 Report
healthcare-2448883_2review
Modified Title: Physical Exercise and Occupational Therapy at Home to Improve the Quality of Life in Subjects Affect by Rheumatoid Arthritis: a randomized controlled trial
Comments for Authors
Dear authors,
I was glad to have the opportunity to review the new version of your manuscript, which investigated the effects of a home intervention of joint economy with physical exercise and occupational therapy, on the quality of life of subjects with rheumatoid arthritis applied over a period of one month.
In my opinion, you have partially responded to the suggestions for improvement made, you have expanded the information required in the introduction and reformulated the hypotheses and conclusions, I congratulate you for your effort and the work you have done. However, I found some issues in the methodology and results sections that you have not answered and I suggest that be addressed to improve the the paper.
Specific comments:
Abstract
- Page 1, line 32. I suggest you add the full term of ers and crp.
Introduction
- Page 2, line 76. Please, add the necessary references that support the information shown in this paragraph
Methods
You have added in the title that this study is a randomized controlled trial. In that case, have you registered this study in a clinical trials registry such as the Clinical Trail gov? This is essential if it is a clinical trial
- Page 2, line 96. Please, could you add information about the period of time in which the data were recorded? How long did it take to complete the entire assessment?
- Pages 3-4, lines 140-166. The information that appears in the methodology section is disordered; it makes difficult read the manuscript. I suggest you rearrange it. You could add a title to introduce the analysed variables. First explain the variables, that appears in the lines 140-166, and later the intervention with the table 1.
- Page 4, lines 161-162. How could you control that the patients performed the exercises properly?
Results
- Page 5, line 195, This infomrtion is about table 2, please chekt it.
- Page 6, line 206. This infomration is about tables 3 and 4, please check it.
In my opinion, especially since it is a clinical trial, the main purpose is to determine the effectiveness of the intervention. I therefore consider that it is essential to add the comparative analysis between the groups to determine the effect of the intervention on them. Add the comparative analysis of the means between the pre and post between the groups, this is really what can determine if the intervention has had statistically significant effects. I believe that this information is crucial for your manuscript and could enrich it considerably.
- Discussion
- Page 9, line 306. ADLs are well known terms, to make the manuscript easier to read for people who may not be familiar with them add the full terms.
- Page 9, line 326. As I have mentioned before, you do not provide results of the comparison between the groups, therefore you cannot affirm that the improvement that occurs in this parameter is significant or not.
Minor editing of English language required
Author Response
Once again, thank you for your comments helpful to improve the quality of our manuscript.
Abstract
- Page 1, line 32. I suggest you add the full term of ers and crp.
According to your suggestion we added the full term.
Introduction
- Page 2, line 76. Please, add the necessary references that support the information shown in this paragraph
According to your suggestion we added the necessary references.
Methods
You have added in the title that this study is a randomized controlled trial. In that case, have you registered this study in a clinical trials registry such as the Clinical Trail gov? This is essential if it is a clinical trial
We agree with your comment. The trial was approved by the Ethical Committee of our institution. We have not recorded our study because although it is an intervention study it does not deal with drugs.
- Page 2, line 96. Please, could you add information about the period of time in which the data were recorded? How long did it take to complete the entire assessment?
According to your suggestion we added the necessary information about the period of time in which the data were recorded and the entire assessment.
- Pages 3-4, lines 140-166. The information that appears in the methodology section is disordered; it makes difficult read the manuscript. I suggest you rearrange it. You could add a title to introduce the analysed variables. First explain the variables, that appears in the lines 140-166, and later the intervention with the table 1.
According to your suggestion we rearranged the methodology section
- Page 4, lines 161-162. How could you control that the patients performed the exercises properly?
Thank you for your observation. This is out of the aim of the study. We underlined that the intervention is an educational program evaluated by an intention to treat analysis.
Results
- Page 5, line 195, This infomrtion is about table 2, please chekt it.
Thank you for your observation. We corrected the error.
- Page 6, line 206. This infomration is about tables 3 and 4, please check it.
Thank you for your observation. We corrected the error.
In my opinion, especially since it is a clinical trial, the main purpose is to determine the effectiveness of the intervention. I therefore consider that it is essential to add the comparative analysis between the groups to determine the effect of the intervention on them. Add the comparative analysis of the means between the pre and post between the groups, this is really what can determine if the intervention has had statistically significant effects. I believe that this information is crucial for your manuscript and could enrich it considerably.
According to your suggestion we added the comparison analysis between the pre and post intervention groups (see tables)
- Discussion
- Page 9, line 306. ADLs are well known terms, to make the manuscript easier to read for people who may not be familiar with them add the full terms.
According to your suggestion we added the full term-
- Page 9, line 326. As I have mentioned before, you do not provide results of the comparison between the groups, therefore you cannot affirm that the improvement that occurs in this parameter is significant or not.
According to your suggestion we added in the results section the comparative analysis of the means between the pre and post-intervention amidst the groups
